# Collision-Free Robot Path Planning by Integrating DRL with Noise Layers and MPC

**DOI:** 10.3390/s25206263

**Published:** 2025-10-10

**Authors:** Xinzhan Hong, Qieshi Zhang, Yexing Yang, Tianqi Zhao, Zhenyu Xu, Tichao Wang, Jing Ji

**Affiliations:** 1Hangzhou Institute of Technology, Xidian University, Hangzhou 311231, China; 22041212676@stu.xidian.edu.cn (X.H.); 22041212673@stu.xidian.edu.cn (Y.Y.); 22041212754@stu.xidian.edu.cn (T.Z.); 2School of Information Mechanics and Sensing Engineering, Xidian University, Xi’an 710071, China; 3CAS Key Laboratory of Human–Machine Intelligence-Synergy Systems, Shenzhen Institutes of Advanced Technology, Chinese Academy of Sciences, Shenzhen 518055, China; qs.zhang@siat.ac.cn (Q.Z.); tc.wang@siat.ac.cn (T.W.); 4Faculty of Science and Technology, University of Macau, Macau 999078, China

**Keywords:** mobile robots, path planning, deep reinforcement learning, MPC

## Abstract

With the rapid advancement of Autonomous Mobile Robots (AMRs) in industrial automation and intelligent logistics, achieving efficient and safe path planning in dynamic environments has become a critical challenge. These environments require robots to perceive complex scenarios and adapt their motion strategies accordingly, often under real-time constraints. Existing methods frequently struggle to balance efficiency, responsiveness, and safety, especially in the presence of continuously changing dynamic obstacles. While Model Predictive Control (MPC) and Deep Reinforcement Learning (DRL) have each shown promise in this domain, they also face limitations when applied individually—such as high computational demands or insufficient environmental exploration. To address these challenges, we propose a hybrid path planning framework that integrates an optimized DRL algorithm with MPC. We replace the Actor’s output with a learnable noisy linear layer whose mean and scale parameters are optimized jointly with the policy via backpropagation, thereby enhancing exploration while preserving training stability. TD3 produces stepwise control commands that evolve into a short-horizon reference trajectory, while MPC refines this trajectory through constraint-aware optimization to ensure timely obstacle avoidance. This complementary process combines TD3′s learning-based adaptability with MPC’s reliable local feasibility. Extensive experiments conducted in environments with varying obstacle dynamics and densities demonstrate that the proposed method significantly improves obstacle avoidance success rate, trajectory smoothness, and path accuracy compared to traditional MPC, standalone DRL, and other hybrid approaches, offering a robust and efficient solution for autonomous navigation in complex scenarios.

## 1. Introduction

With the widespread application of Autonomous Mobile Robots (AMRs) in fields such as industrial automation and intelligent logistics, achieving efficient and collision-free path planning and obstacle avoidance in dynamic environments has become a critical research topic [1,2,3]. Particularly in complex dynamic environments, robots must not only avoid static obstacles but also deal with constantly changing dynamic obstacles, such as pedestrians and other robots. Existing path planning methods perform well in static environments, but they often fail to effectively guarantee real-time performance and safety when facing dynamic obstacles.

Model Predictive Control (MPC) [4], as a widely used path planning and obstacle avoidance method, is especially suitable for local planning in dynamic environments. MPC optimizes control inputs at each time step to minimize the objective function while satisfying various constraints (such as obstacle avoidance and velocity limits). This allows real-time adjustments to the control strategy to cope with dynamic changes in the environment, particularly excelling in obstacle avoidance tasks. However, although MPC can provide precise control within a local range, it still faces challenges in global path planning, particularly when dealing with complex non-convex obstacles and high computational complexity, which may lead to low computational efficiency and slow responses to environmental changes. Therefore, relying solely on MPC for global planning may not meet the requirements for real-time performance and computational efficiency, often necessitating its combination with other algorithms to enhance global optimization capabilities [4,5,6,7].

In recent years, Deep Reinforcement Learning (DRL) [8] has been introduced into path planning and obstacle avoidance tasks, showing significant advantages, especially in high-dimensional state spaces and complex action control scenarios. For example, improved versions of DRL algorithms such as Deep Q-Network (DQN) [9], Deep Deterministic Policy Gradient (DDPG) [10], and Twin Delayed Deep Deterministic Policy Gradient (TD3) [11] have been designed with specific mechanisms to enhance exploration to some extent and to actively address local optima, achieving some success. However, DRL still faces the problem of insufficient exploration in complex dynamic environments. Especially in complex scenarios, models tend to get stuck in local optima, limiting the efficiency and precision of path planning and obstacle avoidance tasks, which requires further optimization [12,13,14].

To address these challenges, this paper proposes a collaborative optimization strategy that integrates the TD3 model with MPC, as shown in Figure 1**.** In this approach, TD3 produces sequential local actions that roll out into a short-horizon reference path, which MPC continuously refines for real-time local adjustments. A dynamic weight adjuster fine-tunes the trajectory to enhance obstacle avoidance and improve navigation efficiency.

Specifically, we propose an optimization scheme that replaces the Actor output layer in the TD3 network with a learnable noise layer, thereby enhancing exploration capability while maintaining training stability. The optimized TD3 is then integrated with MPC: TD3 produces sequential local actions that roll out into a short-horizon reference path, while MPC refines this path through constraint-aware local optimization for real-time obstacle avoidance. This collaboration improves both path planning accuracy and obstacle avoidance capability. Extensive experimental validation demonstrates that the proposed method outperforms traditional MPC, standalone DRL algorithms, and other hybrid optimization approaches in static and dynamic environments, achieving higher success rates, smoother trajectories, and more accurate paths. The main contributions of this paper include:

(1) A TD3-based path planning framework enhanced with an embedded noise layer in the Actor network is proposed, which improves exploration capability and convergence stability compared to conventional action noise injection.

(2) The TD3 planner is integrated with an MPC-based local controller through a dynamic weight-switching mechanism, where TD3 provides sequential local actions that roll out into a reference path, and MPC refines these actions for responsive local obstacle avoidance.

(3) The proposed framework is designed for and evaluated with fused onboard sensor data, combining 360° LiDAR, odometry, and IMU measurements. This sensing configuration enables robust operation in realistic, highly dynamic environments and differs from previous approaches that rely solely on fixed-view visual inputs.

(4) Extensive experiments in static and dynamic scenarios demonstrate superior success rate, path smoothness, and safety margins compared to pure MPC, pure RL, and prior DRL+MPC baselines.

## 2. Related Works

### 2.1. MPC Based Path Planning

MPC, as an optimization-based path planning method, is widely applied in robotic path planning and obstacle avoidance tasks. Eckhoff et al. [15] innovatively proposed the application of MPC in safe Human–Robot Interaction (HRI) motion planning. By optimizing and predicting future states based on constrained optimization, it actively avoids robot motion safety conflicts, such as human–robot collisions, overcoming the limitations of traditional control schemes that react only during motion conflicts. This approach provides a proactive path planning strategy for safe HRI. Ramezani et al. [16] combined MPC with Long Short-Term Memory (LSTM) networks to address the unmanned aerial vehicle path planning problem in complex environments. Zhao et al. [17] proposed a solution combining an improved A* algorithm and MPC, where MPC outputs control signals based on vehicle dynamics models, integrating predictive optimization and custom penalty functions to precisely handle complex parking scenarios, effectively improving the quality and control accuracy of parking trajectories.

However, the application of MPC in complex dynamic environments still faces challenges. Especially in high-computation-demand and uncertain environments, MPC often falls into local optima, and its high computational complexity results in poor real-time performance. These limitations make it difficult for MPC to cope with rapidly changing dynamic environments when used alone. To address this, researchers have started exploring the combination of MPC with other intelligent algorithms to enhance its adaptability and real-time responsiveness in dynamic environments.

### 2.2. DRL Based Path Planning

In recent years, DRL has been widely applied in robotic path planning and control, especially in complex tasks involving high-dimensional state spaces and continuous action spaces, where it shows significant advantages. Yan et al. [18] proposed a DRL-based UAV path planning method, using a situation assessment model and the Duel Deep Q-Network (D3QN) algorithm to predict Q-values and select actions based on specific policies, providing a new approach for DRL in path planning. Li et al. [19] used the TD3 algorithm for mobile robot path planning, introducing Prioritized Experience Replay and Transfer Learning to improve learning efficiency. They also designed a dynamic delay update strategy and incorporated Ornstein-Uhlenbeck (OU) noise, improving the success rate of path planning by 16.6%. Han et al. [20] introduced a novel perspective by designing differentiable deterministic factors, proposing denoising methods, and constructing an online weight adjustment mechanism, offering new insights into the incorporation of noise layer into DRL algorithms and opening new avenues for optimizing noise layer.

### 2.3. MPC and DRL Combined Path Planning

In reference [21], the authors proposed a hybrid switching-driven algorithm that integrates MPC and DRL, successfully applying it to planar non-grasping operations for robots. This approach significantly enhanced both the training and execution efficiency of the original system. Zhang et al. innovatively combined DRL with MPC by training DQN variants tailored to different observations, optimizing action selection, and designing new reward functions and switching strategies [22]. Their experiments across multiple scenarios demonstrated that the proposed hybrid method outperformed standalone MPC and DQN algorithms in various performance metrics, offering an efficient solution to robot navigation challenges. Later, Zhang et al. introduced a hybrid method combining bird’s-eye-view vision-based DRL with MPC to solve complex trajectory generation and obstacle avoidance problems [23]. The study involved designing a DDPG model, optimizing hybrid strategies, and performing evaluations across multiple scenarios, which revealed superior performance in robot settings, particularly in terms of new scene adaptation, computational efficiency, obstacle avoidance, and collaborative performance, when compared to pure DRL or MPC methods. However, despite the high success rate of model tests in static obstacle environments and occasional tests with dynamic obstacles, the motion trajectories of obstacles remained relatively fixed. This suggests there is still potential for improvement, particularly in handling randomly dynamic obstacles and more complex, evolving environments.

## 3. Methodology

### 3.1. Problem Definition

In dynamic and complex environments, robot trajectory planning requires effective coordination between global path planning and real-time local optimization while addressing challenges such as dynamic obstacles and diverse scenarios. Figure 2 illustrates the proposed framework, which integrates TD3 with MPC to balance global decision-making and local adjustments. TD3 employs a learnable noise layer design to enhance exploration capabilities and robustness, outputting sequential local actions that, when rolled out, form a short-horizon reference path while optimizing long-term cumulative rewards. Meanwhile, MPC incorporates dynamic obstacle prediction and kinematic constraints to perform real-time local trajectory optimization and dynamic obstacle avoidance. The two components collaborate adaptively through a dynamic weight adjustment module, which allocates control strategy weights based on obstacle density and environmental complexity. This synergy enables the generation of smooth, safe, and efficient robot control commands. The framework achieves closed-loop optimization at both global and local levels, providing an efficient and robust solution for robot navigation in challenging environments.

#### 3.1.1. Kinematic Modeling

In robot path planning, the kinematic model is critical for ensuring that the robot can move efficiently while avoiding obstacles in dynamic environments. The non-holonomic differential drive model is employed in this study, which represents the robot’s motion in terms of its position, orientation, and velocities. The model provides a simple yet effective representation of a mobile robot’s motion and is particularly suitable for systems with two wheels, like differential drive robots, which are commonly used in practical applications. The discrete-time form of the kinematic model is given by the following equations:(1)$$\begin{array}{c} x_{t+1}=x_{t}+v_{t}\cdot \cos\left(\theta_{t}\right)\cdot ∆t, \end{array}$$(2)$$\begin{array}{c} y_{t+1}=y_{t}+v_{t}\cdot \sin\left(\theta_{t}\right)\cdot ∆t, \end{array}$$(3)$$\begin{array}{c} \theta_{t}=\theta_{t}+\omega_{t}\cdot ∆t, \end{array}$$

Here, $x_{t},y_{t},\theta_{t}$ represents the robot’s position and orientation at time *t*, $v_{t}$ and $\omega_{t}$ are the linear and angular velocities of the robot, respectively; and ∆t is the time step.

The robot’s trajectory planning depends heavily on the accurate modeling of its movement, which can be constrained by real-world limitations such as the robot’s maximum velocity and acceleration. To ensure controllability and stability in the robot’s motion, the following kinematic constraints are applied:

Linear velocity and acceleration: The linear velocity is constrained within the range [−0.5,1.5] m/s, where negative values represent allowable reverse motion. To avoid instability caused by sudden acceleration or deceleration, the linear acceleration is restricted within [−1.0,1.0] m/s^2^, thus controlling the rate of change in velocity.

Angular velocity and acceleration: To control the amplitude of the robot’s rotational motion, the angular velocity is constrained within [−0.5,0.5] rad/s. The range of angular acceleration is defined as [−3.0,3.0] rad/s^2^, in order to ensure smooth rotational motion and avoid oscillations.

Discretization and Sampling Time: To implement the above constraints in a digital control system, the kinematic model is discretized with a sampling time of ∆t = 0.2 s. This discretized time step ensures that the robot can respond promptly to changes in the dynamic environment during planning and is consistent with the control step size in the MPC optimization framework.

The application of these kinematic constraints ensures that the robot can operate within safe and practical limits. They are essential when integrating the robot’s kinematic model with the global path planning and local optimization components, particularly in the context of dynamic environments where obstacle densities and environmental conditions constantly change. This modeling approach helps maintain a balance where TD3 outputs local control actions that roll out into a reference path, while MPC performs local optimization for real-time obstacle avoidance, which are critical for the robot’s ability to navigate complex, dynamic scenarios.

#### 3.1.2. Optimization Objective Modeling

To achieve effective robot navigation in dynamic and complex environments, the trajectory planning task is framed as an optimization problem. The primary objective is to generate a collision-free trajectory from the start point to the target, while maintaining smoothness, efficiency, and control input feasibility. The optimization cost function is designed to combine several critical factors:(4)$$J=\sum_{t=0}^{N}\left(w_{1}\cdot J_{path}+w_{2}\cdot J_{smooth}+w_{3}\cdot J_{collision}+w_{4}\cdot J_{control}\right),$$
where path tracking cost ($J_{path}$) measures the deviation between the robot’s actual trajectory and the reference path, which is critical for accurate path following:(5)$$J_{path}=||s_{t}-s_{ref}{\left|\right|}^{2},$$

Here, $s_{t}=(x_{t},y_{t})$ is the robot’s current position and *s_ref_* is the reference trajectory point.

Trajectory smoothness cost ($J_{smooth}$) limits abrupt turns or accelerations during trajectory generation:(6)$$J_{smooth}=||u_{t}-u_{t-1}{\left|\right|}^{2},$$
where $u_{t}=(v_{t},w_{t})$ represents the control inputs, including linear and angular velocities.

Collision cost ($J_{collision}$) is used to avoid collisions between the robot and static or dynamic obstacles:(7)$$\begin{array}{c} J_{collision}=\sum_{o\in O}\max\left(0,d_{safe}-||s_{t}-o_{t}{\left|\right|}^{2}\right), \end{array}$$

Here, $o_{t}=(x_{o},y_{o})$ represents the position of the obstacle at time *t*, O is the set of all obstacles in the environment, and $d_{safe}$ is the predefined safe distance.

Control cost ($J_{control}$) limits the magnitude of control inputs, ensuring that the robot operates within its physical capabilities:(8)$$\begin{array}{c} J_{control}=v_{t}^{2}+w_{t}^{2}. \end{array}$$

The weight coefficients $\left(w_{1},w_{2},w_{3},w_{4}\right)$ in the cost function can be adjusted according to task requirements to balance path tracking and control overhead.

#### 3.1.3. Sensing and Observation

In this work, the observation state $s_{t}$ is derived from onboard multi-sensor data fusion rather than a top-mounted Bird’s-Eye-View (BEV) camera. A 2D LiDAR sensor provides 720 evenly spaced range measurements over 360°, normalized to [0,1] using the sensor’s maximum range of 3.5 m, and transformed into a local occupancy grid aligned with the robot’s current heading. This grid is processed by a lightweight Convolutional Neural Network (CNN) to extract spatial features. These features are combined with proprioceptive information, including linear and angular velocities from wheel odometry, and three-axis angular velocity and linear acceleration from an Inertial Measurement Unit (IMU). In the real-world experiments, a forward-facing RGB camera is optionally used to provide supplementary visual cues for detecting obstacle boundaries and environmental details; however, it is not included in the DRL training process.

The action vector $a_{t}$ is two-dimensional, consisting of the linear velocity ν and angular velocity ω for the differential-drive base. The ranges are constrained to v∈[0,0.22] m/s and ω∈[−2.84,2.84] rad/s to match the physical limits of the robot platform. This sensing-action configuration supports robust operation in both simulated and real-world environments with stochastic dynamic obstacles, without relying on top-mounted BEV cameras [23].

#### 3.1.4. Dynamic Obstacle Prediction Modeling

In dynamic environments, the random motion of obstacles increases the difficulty of path planning. Therefore, this paper designs a kinematic-based obstacle trajectory prediction model to update dynamic constraints in real time during the optimization process. The motion model of an obstacle is given by:(9)$$\begin{array}{c} o_{t+1}=o_{t}+∆t\cdot v_{o,} \end{array}$$
where $o_{t}=(x_{o},y_{o})$ represents the obstacle’s position and $v_{o}=(v_{x},v_{y})$ represents its velocity. Additionally, to enhance the robustness of the prediction, a combination of short-term trajectory regression and motion trend analysis is employed, and a sliding window is used to smooth the obstacle trajectory:(10)$$\begin{array}{c} o_{t+1}=\sigma \cdot o_{t}+\left(1-\alpha \right)\cdot \left(o_{t-1}+v_{o}\cdot \Delta t\right), \end{array}$$
where *α* ∈ [0,1] is the smoothing factor. The predicted obstacle trajectories are used to dynamically update the obstacle constraints in the optimization problem, enabling the MPC module to effectively avoid dynamic obstacles.

### 3.2. TD3 Model with Noise Layer Optimization

Traditional TD3 implementations typically inject i.i.d. Gaussian noise directly into the action space to promote exploration. However, such post hoc perturbations do not influence the policy parameters during backpropagation, and therefore cannot structurally shape how the policy improves. This limits their ability to provide consistent, gradient-informed exploration.

To overcome this limitation, we redesign the Actor’s output layer as a learnable noisy linear layer inspired by parameter-space noise. Instead of applying external noise after the action is computed, the perturbation is embedded into the weight and bias parameters of the output layer and updated together with the Actor through backpropagation. This mechanism produces state-consistent, rollout-level exploration that directly impacts the replay buffer distribution while preserving the stability of TD3 through a noise-free twin-Critic. In this way, exploration is structurally coupled with learning, which is fundamentally different from conventional post hoc action noise. In our final design, noise injection is applied only to the Actor network to avoid destabilizing Critic value estimation.

During data collection, the noisy layer is enabled, and actions are sampled with stochastic perturbations, generating rollout-consistent exploration that populates the replay buffer. For target policy evaluation, the noisy layer is disabled and its deterministic expectation is used, preserving the stability of TD3′s target update. During Actor updates, we employ a single-sample reparameterization strategy with the sampled perturbation fixed per minibatch, which allows gradients to flow into the noise parameters while keeping variance under control. This design ensures that exploration directly influences the training data while maintaining the deterministic backbone required for stable policy optimization.

#### 3.2.1. Noise Layer Design Principles

The noise layer is a mechanism that injects parameterized randomness into the network during training, promoting exploration by introducing stochasticity into the policy, especially in high-dimensional continuous action spaces [24]. The core design consists of the following three components:

(1) Randomization of weights and bias parameters

Figure 3 compares a standard linear layer with a noise layer, highlighting the parameterized randomness design of the noise layer. The linear transformation of each noise layer is redefined as:(11)$$\begin{array}{c} y=\left(\mu_{\omega }+\sigma_{\omega }⨀\epsilon_{\omega }\right)x+\left(\mu_{b}+\sigma_{b}⨀\epsilon_{b}\right), \end{array}$$
where $\mu_{\omega }$ and $\mu_{b}$ are learnable means values of the weight and bias, respectively, acting as deterministic parameters. $\sigma_{\omega }$ and $\sigma_{b}$ are learnable scales (all registered as trainable tensors), updated jointly with the Actor via backpropagation. $\epsilon_{w}$ and $\epsilon_{b}$ are noise variables drawn from a standard normal distribution *N*(0,1). This design dynamically adjusts the amplitude of noise injection, facilitating a balance between exploration and exploitation during different training stages.

(2) Noise decay mechanism

To mitigate over-exploration in later stages without overriding the learnable scale parameters $\sigma_{\omega }$ and $\sigma_{b}$, we apply a weak annealing coefficient only to the sampled perturbations. The specific formula is:(12)$$\begin{array}{c} \tilde{\varepsilon_{t}}=\alpha_{t}\varepsilon ,\;\alpha_{t}=\alpha_{0}\exp\left(-\frac{t}{\tau }\right),\;\varepsilon \sim N\left(0,1\right). \end{array}$$

This annealing biases the effective injected noise toward smaller values over time, while $\sigma_{\omega }$ and $\sigma_{b}$ remain fully trainable and continue to be optimized by backpropagation. In this way, exploration stays strong early on and naturally attenuates as learning progresses, without hard-coding a schedule onto the learnable scales.

(3) Noise smoothing and stability enhancement

To address the issue of noise fluctuations potentially affecting model convergence, an Exponential Moving Average (EMA) smoothing mechanism is introduced, with the following formula:(13)$$\begin{array}{c} \epsilon_{\omega }^{\left(t\right)}=\sigma \epsilon_{\omega }^{\left(t-1\right)}+\left(1-\sigma \right)\epsilon_{\omega }^{\left(t\right)}, \end{array}$$
where *σ* is the EMA decay factor that controls the degree of smoothing. The introduction of EMA dynamically smooths the noise by giving higher weight to the historical noise, effectively reducing the severity of noise fluctuations. This smoothing process not only reduces the interference of noise on policy generation but also enhances the model’s stability and convergence efficiency during training, ensuring robust exploration while avoiding excessive noise interference in model optimization. EMA operates on sampled perturbations; the underlying *μ* and *σ* parameters remain fully learnable.

#### 3.2.2. TD3 Network Structure with Noise Layer

To improve the exploration capability and convergence stability of the TD3 in dynamic environments, this study integrates a noise layer into the output layer of the actor network while retaining the dual Q-network architecture in the Critic network [25], forming the overall framework shown in Figure 4.

The Actor network extracts high-dimensional features of the environment state $s_{t}$ through a feature extraction module (Conv2D + ReLU) and passes them through two hidden layers (each containing 64 neurons) to generate the control action $a_{t}$. To enhance the diversity of policy generation, the output layer is redefined as a learnable noisy linear layer, where both the mean and scale parameters are optimized via backpropagation together with the Actor. The Tanh activation function is applied to constrain the action values within the range [−1,1]. The noisy output layer injects perturbations through trainable mean and scale parameters; under gradient pressure (and the weak decay prior), the learned variance gradually decreases, yielding stable policies in later training.

The Critic network evaluates the value of the action $a_{t}$ generated by the Actor using the dual Q-network structure, which computes two independent Q-values, Q1 and Q2, to address the problem of Q-value overestimation. In reinforcement learning, the Q-value represents the expected future rewards for a given state-action pair. However, using a single Q-value can lead to overestimation, particularly when the value function is approximated from noisy or biased data. To mitigate this, the dual Q-network structure independently estimates the Q-value through Q1 and Q2. The final Q-value used for learning is typically selected as the smaller of the two estimates, helping to prevent overestimation and promoting more stable learning.

These Q-values are computed after concatenating the state $s_{t}$ and action $a_{t}$, which are processed through fully connected layers (each with 64 neurons). The state features, including both external aspects (e.g., visual information) and internal factors (e.g., speed and direction), are combined multidimensionally to improve the network’s understanding of the environment. This holistic representation of the state enhances the ability of the Critic network to evaluate the action’s value accurately.

Through this design, the integration of the noise layer in the Actor network enhances the exploration capability of the policy network while not interfering with the Q-value evaluation in the Critic network. This provides greater robustness and efficiency for policy optimization in dynamic environments.

### 3.3. Collaborative Optimization Mechanism of TD3 and MPC

In dynamic environments, robot path planning involves balancing global planning and local real-time control. The collaboration between TD3 and MPC efficiently integrates both aspects, improving robot navigation performance.

#### 3.3.1. TD3 Provides Short-Horizon Action Rollouts

The Actor network outputs stepwise velocity commands; rolling them out over the control horizon yields a short-horizon reference path that conditions MPC’s local optimization. This reference is then supplied to the MPC module, which adapts it to the current environment for locally feasible execution.

#### 3.3.2. MPC Performs Local Optimization Based on the Short-Horizon Reference

MPC takes the provisional path obtained from TD3′s action rollouts and further optimizes it under kinematic constraints and predicted obstacle motion, producing control commands that are both safe and dynamically feasible. The optimization objective function for MPC is defined as:(14)$$J=\sum_{t=0}^{N}(q_{r}||x_{t}-x_{ref}{\left|\right|}^{2}+q_{\mu }||\mu_{t}{\left|\right|}^{2}+q_{obs}\sum_{i=1}^{M}dist(x_{t},o_{i})),$$
where $x_{t}$ is the robot’s current state, $x_{ref}$ is the reference trajectory generated by TD3, $u_{t}$ is the control input, and $o_{i}$ represents the positions of dynamic obstacles. The parameters $q_{r},q_{u},q_{obs}$ are the weights for the reference trajectory deviation, control input cost, and obstacle distance cost, respectively.

By optimizing this objective function, MPC generates smooth and safe local trajectories in dynamic environments, avoiding collisions with obstacles while enhancing the smoothness and robustness of local control [26,27]. The local optimization process of MPC effectively adapts the short-horizon candidate path to the dynamic environment, enabling quick responses to environmental changes and improving reliability.

#### 3.3.3. Dynamic Weight Adjustment for Collaborative Optimization

To coordinate the TD3 planner and MPC controller, a logistic switching function is introduced to dynamically adjust their relative contributions according to the observed obstacle density:(15)$$\begin{array}{c} \lambda_{\left(t\right)}=\frac{1}{1+exp(-k\left(d_{t}-d_{c}\right))}, \end{array}$$
where $d_{t}$ denotes the obstacle density at time step t, $d_{c}$ is a predefined threshold, and *k* determines the steepness of the transition curve. When $d_{t}>d_{c}$, the value of $\lambda_{\left(t\right)}$ increases, giving MPC greater control authority to ensure collision avoidance; otherwise, TD3 retains a larger share to leverage its long-horizon planning capability.

In this work, $d_{c}=0.5$ and k=5 were determined through preliminary simulation trials, balancing responsiveness and stability while avoiding oscillatory switching between controllers. Empirical observations indicate that moderate variations (±20%) in these parameters have negligible effects on success rate and trajectory smoothness, suggesting that the method is not highly sensitive to exact parameter values. This mechanism ensures smooth and adaptive transitions between global and local control without manual intervention.

## 4. Experiments

The effectiveness of the noise layer replacement scheme in the TD3+MPC model is validated through experiments, focusing on the impact of different noise replacement positions on model performance and comparing it with traditional algorithms. The results are presented through reward curves, map scene evaluations, and performance data tables, highlighting the superiority and practical applicability of the proposed design.

### 4.1. Experimental Setup

The experiments were conducted on a computing platform with a GPU (NVIDIA RTX 4060) and CPU (Intel i9-12900). The TD3 network received fused onboard sensor data as its state input, combining 360° LiDAR range measurements with odometry and inertial sensor readings to provide both obstacle proximity and motion state information. This setup reflects a realistic onboard sensing configuration for mobile robots, differing from prior works [22,23] that rely on overhead or fixed-view camera data. The action space consisted of continuous control over speed and direction. The hyperparameters used in the training process are shown in Table 1.

These parameters ensured sufficient training and stability for the model. During the training process, multiple parallel environments were used to accelerate data sampling, and moderate policy noise and a discount factor *γ* were introduced to balance exploration and long-term rewards effectively. These hyperparameters were chosen based on several experimental validations, forming a reliable foundation for subsequent comparative studies. Note that the ‘policy noise’ in Table 1 refers to TD3′s target policy smoothing for the target action (as in the canonical TD3), not to an additional exploration noise at action-space; exploration is primarily provided by the learnable parameter-space noisy output layer, whose mean and scale parameters are optimized via backpropagation while sampled noise drives exploration during data collection.

### 4.2. Reward Curve Comparison: Different Noise Layer Replacement Schemes

The primary goal of the noise-layer integration is to enhance exploration during policy learning while ensuring stable convergence of the model. Figure 5 presents representative reward curves for four noise-layer configurations. Each curve shows the smoothed average reward obtained from periodic evaluation rollouts conducted during training, illustrating the general learning dynamics and convergence tendencies. These curves are intended to highlight relative trends, while large-scale statistical robustness is examined through the scenario-based evaluations in Section 4.3.

The baseline TD3 model without noise integration (Figure 5a) converges locally around a reward value of 70. Replacing the Actor output layer with a noise layer (Figure 5b) overcomes this premature convergence and yields a substantially higher final reward. Perturbing both the Actor and Critic outputs (Figure 5c), however, reduces stability and results in a lower final reward, confirming that noise at the Critic side injects variance into Q-value estimation and hinders convergence. Similarly, perturbing both the Actor output and a hidden layer (Figure 5d) initially boosts exploration but later causes sharp fluctuations and even declines, indicating that excessive noise interference outweighs any transient benefit.

These findings support the decision to prioritize the Actor network for noise layer replacement. In the TD3 architecture, the Critic’s primary role is to estimate Q-values, while the Actor is responsible for generating policies. Therefore, modifying the Actor network has a more direct impact on exploration and policy optimization than altering the Critic. For similar reasons, this study did not explore noise replacement in the input layer, as it could distort the state feature representation and hinder environmental perception.

Unlike conventional TD3 implementations that inject i.i.d. external action noise, the proposed method embeds a learnable, reparameterized noisy output layer in the Actor. Both mean and scale parameters are optimized by backpropagation, while sampled perturbations are applied during data collection and optionally frozen per minibatch during policy updates to control gradient variance. For target policy evaluation, the deterministic expectation of the noisy layer is used to preserve the stability of TD3. This mechanism integrates exploration directly into the learning pipeline and provides trajectory-level consistency, fundamentally distinguishing it from post hoc action noise.

### 4.3. Algorithm Performance Evaluation: Comparison Across Multiple Map Scenes

To validate the performance of the TD3+MPC algorithm with Actor output layer replacement in various environments, we designed three representative evaluation scenes (Scenes A, B, and C) shown in Figure 6, are more challenging than those used in the evaluation environments of previous studies [23], featuring more dynamic obstacles, more complex obstacle distributions, and higher task difficulty. The aim is to comprehensively test the algorithm’s adaptability in dynamic environments, narrow passages, and dense obstacle scenarios.

The experiments compare the performance of the traditional pure MPC method, the TD3 algorithm, and other deep reinforcement learning-based combinations. The performance evaluation metrics include computation time, action smoothness, minimum safety distance (clearance), task completion time steps, and success rate. Action smoothness is quantified by the mean absolute change in both linear and angular velocities between consecutive time steps, reflecting the stability of robot motion control. Among these, clearance refers to the minimum distance between the robot’s trajectory and the nearest obstacle, reflecting the safety margin of path planning. The results of these evaluations are summarized in Table 2.

In Scene A, representing a static and structured obstacle environment, MPC and MPC+DDPG both maintained full success rates, while MPC+DQN achieved slightly lower stability. TD3 alone performed poorly due to its lack of local optimization. By contrast, MPC+TD3 also achieved a 100% success rate and demonstrated reduced computational burden per step compared with pure MPC. Although its completion steps were somewhat longer than those of MPC and MPC+DQN, the trajectories remained stable and collision-free, indicating that the hybrid method can preserve safety while improving efficiency in controller execution.

In Scene B with randomized dynamic obstacles, MPC+TD3 achieved a higher success rate than all optimization-based baselines, approaching the robustness of standalone TD3 but with significantly improved safety margins. While its action smoothness and completion steps were less favorable, this reflected proactive velocity adjustments and pre-emptive slowdowns in response to obstacle interactions. Such behavior emphasizes that the hybrid method prioritizes collision avoidance and adaptability under uncertainty, ensuring reliable navigation even at the expense of perfect smoothness.

**Table 2 sensors-25-06263-t002:** Evaluation results (average over 100 runs).

Scene	Method	Computation Time (ms/step)	Action Smoothness		Clearance (m)	Finish Time Step	Success Rate (%)
			Speed	Angular			
Scene A	MPC [4]	17.28	0.03	0.01	1.16	312.8	100
	TD3 [28]	3.07	0.50	0.99	0.48	286	2
	MPC+DQN [22]	21.64	0.01	0.03	0.73	130.07	90
	MPC+DDPG [23]	16.91	0.03	0.01	1.15	336.2	100
	MPC+TD3	14.85	0.03	0.05	1.16	330.96	100
Scene B	MPC	17.52	0.03	0.07	0.77	103.55	76
	TD3	1.93	0.10	0.31	1.11	569.79	94
	MPC+DQN	22.84	0.02	0.04	0.74	126.49	80
	MPC+DDPG	18.20	0.04	0.08	0.78	108.36	60
	MPC+TD3	18.13	0.52	0.48	0.74	147.85	87
Scene C	MPC	/	/	/	/	/	0
	TD3	2.14	0.15	0.20	0.45	126.38	16
	MPC+DQN	22.17	0.02	0.03	0.75	132.87	76
	MPC+DDPG	18.45	0.02	0.05	0.72	147.11	90
	MPC+TD3	14.22	0.03	0.06	0.78	107.33	98

Scene C, involving densely cluttered obstacles, posed the most difficult challenge. Here, MPC failed to complete the task, and standalone TD3 exhibited very low stability. MPC+DQN and MPC+DDPG achieved higher success rates, but both suffered from longer completion times and unstable motion patterns. By contrast, MPC+TD3 reached the highest success rate among all tested methods and required noticeably fewer completion steps than MPC+DDPG, while maintaining superior clearance. These results confirm that the hybrid approach can effectively combine TD3′s exploratory capability with MPC’s reactive refinement to produce efficient and safe navigation even in the most complex scenarios.

Overall, MPC+TD3 consistently demonstrated significant advantages across all scenarios. Compared to standalone TD3 or traditional optimization methods, MPC+TD3 integrates the global exploration capabilities of reinforcement learning with the local optimization strength of MPC, exhibiting enhanced adaptability and robustness in complex, dynamic environments. Furthermore, compared to recently proposed methods like MPC+DQN and MPC+DDPG, MPC+TD3 not only achieved higher success rates in challenging scenarios but also outperformed in key metrics such as completion time and trajectory smoothness. These improvements are attributed to the twin-Critic architecture with delayed policy updates in TD3, together with the learnable parameter-space noisy output layer that structurally integrates exploration into the Actor. This synergy enhances strategy generation and ensures both adaptability and stability in complex dynamic environments. In conclusion, MPC+TD3 offers an effective solution for path planning in complex dynamic environments and provides a promising direction for further integration of reinforcement learning with MPC.

### 4.4. Real-World Platform Validation Experiment

To further verify the adaptability and effectiveness of the proposed path planning algorithm in real-world conditions, a series of physical experiments were conducted using a self-assembled mobile robot platform. The platform consists of a four-wheel differential drive chassis, an onboard embedded control unit, a power system, and multiple environmental sensors, enabling scene perception and autonomous decision-making capabilities.

The input state of the TD3 agent consists of 360° LiDAR range data, robot odometry (linear and angular velocity), and IMU readings. These proprioceptive and exteroceptive signals are fused to represent the robot’s state in the policy network. An RGB camera was mounted only for visualization and qualitative analysis during real-world experiments, but it was not used as an input to the proposed algorithm, respectively. In addition, the system integrates IMU and encoder data to achieve accurate pose estimation and real-time feedback. The overall architecture is built on the Robot Operating System (ROS), supporting seamless deployment and online control of the proposed model. The test environment was set up in a structured indoor laboratory space with a smooth floor and various obstacles arranged to simulate representative indoor navigation scenarios. By deploying the proposed TD3+MPC hybrid algorithm with a noise-layer-optimized actor network, the robot was able to autonomously generate navigation trajectories and perform smooth obstacle avoidance during motion.

As shown in Figure 7, the robot autonomously planned its path and successfully navigated through multiple static obstacles, with clear temporal progression captured in each frame. The generated trajectory exhibited strong continuity and smooth transitions, while no collisions occurred throughout the experiment. These results confirm the robustness and practical applicability of the proposed method under real-world deployment conditions.

## 5. Conclusions

This study presented a hybrid optimization framework that integrates TD3 with an output-layer noise mechanism and MPC to address the problem of mobile robot path planning in complex dynamic environments. The embedded noise layer enhances the consistency of exploration during policy learning, while MPC provides real-time local optimization for obstacle avoidance. In this framework, TD3 provides adaptive guidance through learned local actions, and MPC subsequently fine-tunes these actions into executable trajectories, ensuring both long-term adaptability and short-term safety. This division explicitly avoids claiming any global plan from local observations, while preserving long-horizon adaptability through learning and short-term safety via optimization The proposed approach achieved higher success rates, smoother trajectories, and more efficient task completion compared with baseline schemes in static, dynamic, and densely cluttered scenarios. Overall, the incorporation of a learnable output-layer parametric noise layer should be regarded as a practical strategy for enhancing exploration consistency and training stability in TD3. When combined with MPC, the proposed framework demonstrates clear advantages in terms of success rate and safety, particularly in dynamic and densely cluttered environments. While some efficiency or smoothness indicators may be traded off, the method achieves a balanced and reliable performance, particularly under dynamic and densely cluttered conditions where safety and success rate are prioritized over motion smoothness, offering a promising solution for complex real-world navigation tasks.

## Figures and Tables

**Figure 1 sensors-25-06263-f001:**
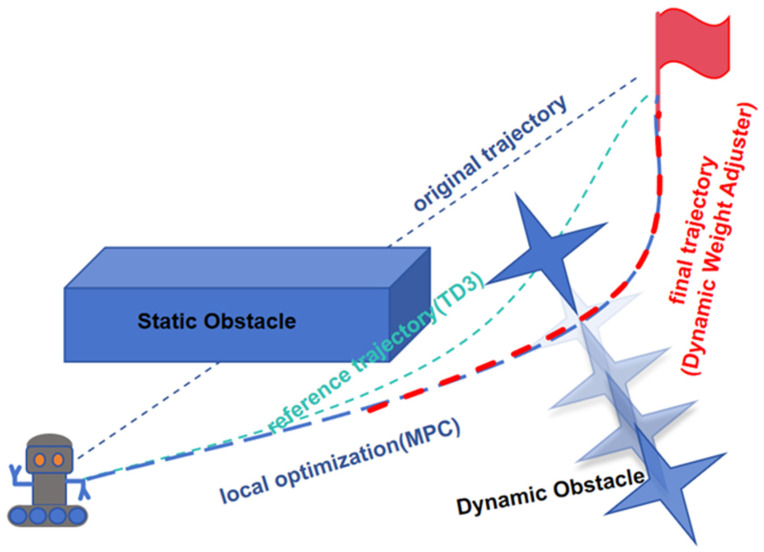
Schematic Path planning process using TD3 and MPC for dynamic obstacle avoidance. The TD3 module provides a sequence of local actions that together approximate a reference path, while MPC continuously adjusts this path in response to environmental changes, ensuring collision-free navigation. The dynamic weight adjuster fine-tunes the final trajectory to avoid both static and dynamic obstacles.

**Figure 2 sensors-25-06263-f002:**
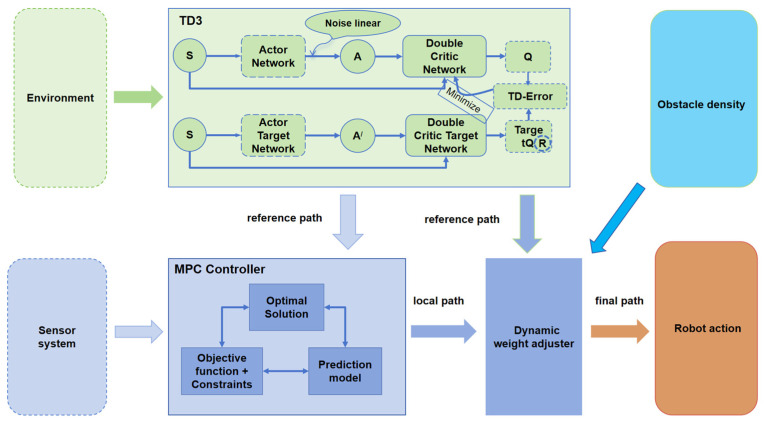
Proposed framework integrating TD3 with MPC for dynamic obstacle avoidance.

**Figure 3 sensors-25-06263-f003:**
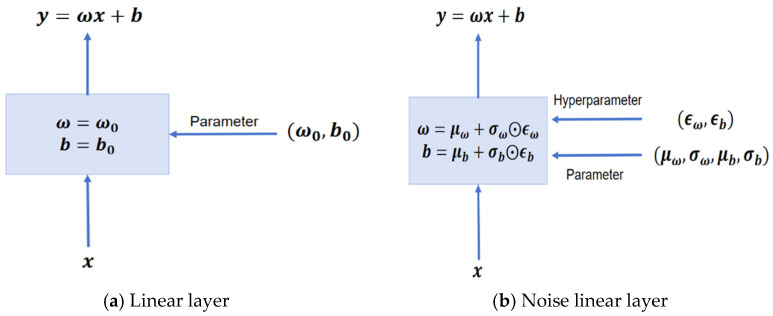
Graphical Representation of Linear Layer and Noisy Linear Layer.

**Figure 4 sensors-25-06263-f004:**
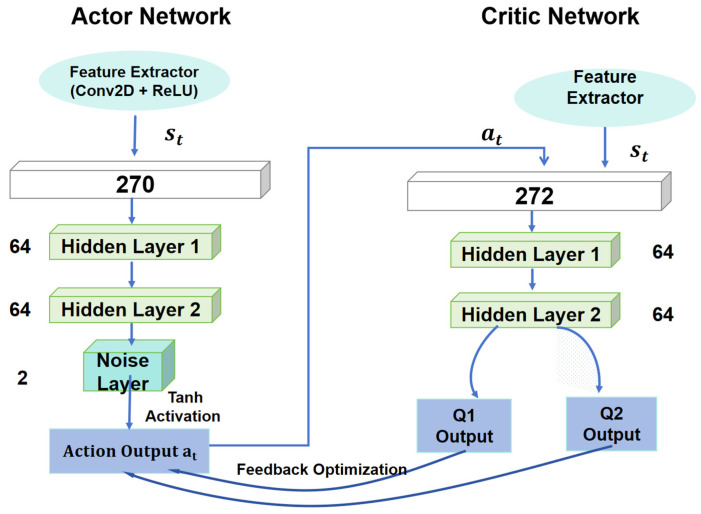
TD3 Network with noise layer.

**Figure 5 sensors-25-06263-f005:**
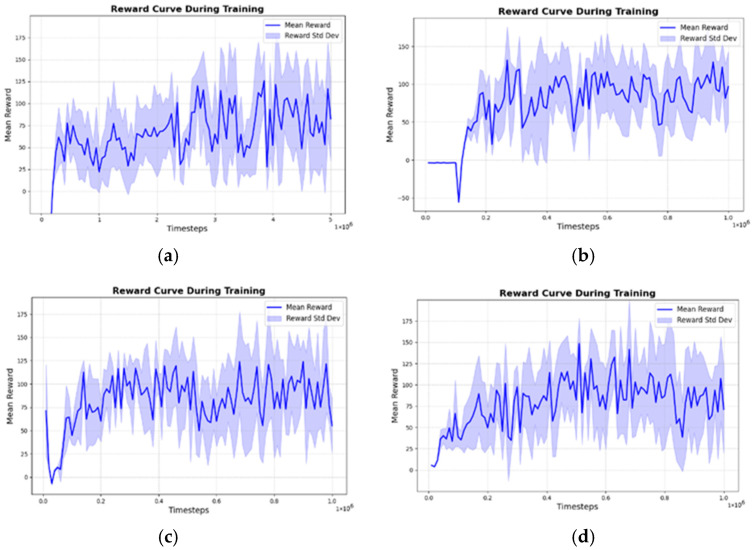
Training reward curves under four noise-layer configurations. Each curve shows the smoothed average reward from periodic evaluation rollouts during training. (**a**) baseline TD3 without noise; (**b**) Actor output with noise layer; (**c**) Actor + Critic outputs with noise layers; (**d**) Actor output + one hidden layer with noise layers.

**Figure 6 sensors-25-06263-f006:**
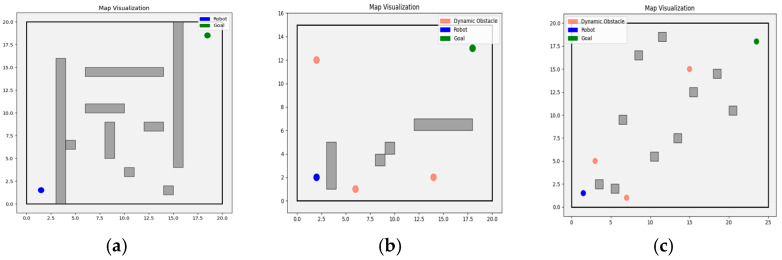
Evaluation Maps: (**a**) Scene A: Static regular obstacle distribution; (**b**) Scene B: Dynamic random obstacle distribution; (**c**) Scene C: High-density random obstacle distribution.

**Figure 7 sensors-25-06263-f007:**
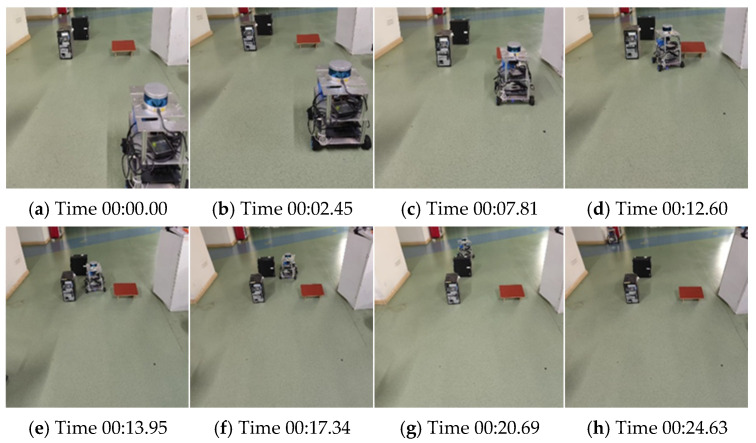
Real-world deployment and validation of the self-assembled mobile robot platform. The time-stamped images illustrate the robot’s real-time navigation and obstacle avoidance process in a structured indoor environment.

**Table 1 sensors-25-06263-t001:** Parameters of the training experiment.

Parameters	Value
Total Timesteps	10^6^
Number of Parallel Environments	8
Discount Factor γ	0.99
Soft Update Factor τ	0.005
Learning Rate	0.01
Replay Buffer Size	10^5^
Batch Size	32
Policy Noise	0.2
Gradient Steps	−1

## Data Availability

Data are available from the corresponding author upon reasonable request.

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
