# Peer review of "Collision-Free Robot Path Planning by Integrating DRL with Noise Layers and MPC"

_sensors, 2025, doi:10.3390/s25206263_

Round 1
Reviewer 1 Report
Comments and Suggestions for Authors
The paper proposes a method that combines DRL based planner with MPC for collision free robot navigation. While the paper style is good and I do not see any language issues, the context is quite confusing and I am not sure if the authors are properly presenting their work and contributions. I find some major issues with the claims in the paper and lack of novelty. With that I am afraid I would have to reject the paper.
- Authors claim the noise layer as a novelty and a key breakthrough in DRL noise design. However, I do not see any major difference to how TD3 algorithms have been implemented until now with regular noise. What is described in the paper is adding noise in an output layer of the model with a certain mean and decaying variance. Controlling the variance controls the level of noise in the model output. But this is no different that having a deterministic output of a model and adding a capped noise afterwards, we have just moved the noise addition in the model. In DRL learning, the Q value estimation and executions will still have the additional noise present thus enabling the exploration. Exactly this is how the exploration is done in most TD3 implementations and is well understood. So for all intents and purposes, simply moving this step inside the model, represents the same addition of noise. While it might be beneficial to have this inside of a model for some reason, this is not explained in the paper and without any additional reasoning I do not see any improvement here for the method. The decaying method is also a rather common decay step calculation. Moreover, the graphs in Fig. 5 are not representative of a performance of any model. First, it is not possible to draw any conclusions for the images themselves, but, second, DRL training is stochastic and as such each individual training run will return a different model. Training each model just once does not truly represent the performance of a model, especially in this case where the differences between the graphs are not visible with a naked eye. Additionally, I fail to see how adding noise to Q network output would positively impact the performance here? Q value is updated with the TD error, then by giving false Q value of the current policy you would give faulty gradients to your parameter update. This is quite confusing and I do not see the purpose of this step.
- It is difficult to see what the novelty in the approach here is. As also mentioned in the paper, similar DRL+MPC methods have been developed in [22] and [23]. Here, the change seemingly is only moving from DDPG algorithm to TD3. While it is implementing a different algorithm, I do not see a major improvement here. TD3 benefit over DDPG is its stability during the training, but other than that, the best DDPG model can work just as well as TD3 method. So if this is the only novelty, I would see it as minor.
- It is really unclear what the model is trained on and what is calculated. What are the inputs to the neural network? What sensors are used? If CNN is used, is it an image? How could the model detect obstacles from that? What is the output of the model? As far as I understand the TD3 model should serve as a reference trajectory, but the Fig. 4 shows that action output is 2 - for diff drive robot that is linear and angular velocities. Then how can the model generate a trajectory? Also as far as I understand TD3 should generate global trajectories? But how can the model do that with only sensor inputs. At least I base that on the text as well as on Fig.1. that also mistakenly refers to the reference trajectory as generated with DQN algorithm. All in all the TD3 method usage needs to be explained in more detail.
- The results are unconvincing. In the 3 scenarios in simulated environments any properly implemented TD3/DDPG/DQN method just from the sensor inputs alone would be able to navigate to the given goal if trained properly. The environments are not that difficult and the low success rate is not consistent to state-of-the-art in similar and more difficult settings.
All in all, I urge the authors to re-evaluate their contribution and differentiate it from existing work.
Author Response
Please see the detailed point-by-point responses in the attached file

Reviewer 2 Report
Comments and Suggestions for Authors
This paper presents a method that combines deep reinforcement learning (DRL, specifically Twin Delayed DDPG TD3) and model predictive control (MPC) to solve the problem of collision-free path planning in dynamic environments. The authors enhanced the TD3 agent with a noise layer on its actor output (along with a decay mechanism) to improve exploration and training stability. In the task, the TD3 agent produces a global trajectory, and the MPC refines it for obstacle avoidance. A dynamic weight adjustment mechanism switches control dominance between the two components based on obstacle density. The authors performed experiments on simulation and observed that the proposed method is effective against pure MPC, pure RL, and prior DRL+MPC hybrids. In the real robot test, the robot successfully navigates without collision.
Strength
- The paper deals with the important robotics problem of real-time planning with moving obstacles, and clearly explains the limitations of existing methods.
- The author listed their contributions without overclaim.
- The ablation studies are comprehensive.
Weakness
- The topic of this paper is RL and control, which might not fit in the topic of the Journal of Advances in Bionic Tactile Systems for Sensor Application and Flexible Operation.
- Please increase the text resolution in the figures to be consistent with the text.
- The central idea of combining DRL with MPC is incremental. There are prior works such as references 22 and 23. And in the proposed method, the authors added the noise layer and heuristic weight-switching mechanism. These are useful, but they are more incremental in nature.
- In Eq.15, a logistic function is used to control the weights, but how are the threshold d_c and k chosen? And how sensitive are they?
- Lines 217 and 242 are both numbered 3.1.2.
- Please define uncommon acronyms at their first use. E.g.. TD3.
- Recommend to mention “Parameter Space Noise for Exploration” (Plappert et al, 2017) in which noise is added to RL parameters instead of actions, and yields consistent exploration.
Author Response

(The authors gave the same response as above.)

Reviewer 3 Report
Comments and Suggestions for Authors
The paper proposes a new methodology for trajectory planning and collision avoidance management of mobile robots. The method is base on the integration of an optimized DRL algorithm with MPC.
On my opinion the method should be described with a deeper detail, in order to allow the reader to repeat the experiments. At the same time, results are discussed only in terms of KPI and success rate, but no actual information about at least a few examples in the considered scenarios is provided.
In order to make the paper suitable for publication the authors should:
1) provide a deeper description of the method and information that can allow the reader to repeat the experiments
2) show the results of at least two case tests for each scenario
Author Response

(The authors gave the same response as above.)

Round 2
Reviewer 1 Report
Comments and Suggestions for Authors
The second round of updates introduces some new questions in the paper and I feel the previous questions were still not addressed to a satisfactory manner.
The biggest issue here is describing the neural network and what the noise layer contributes. The authors argue that introducing the noise layer in the model itself somehow is more beneficial than using adding noise to the output due to backpropagation. However, the adding of the noise is usually attributed to the action and such action is saved in the replay buffer which is used for calculating the loss of critic network. Critic updates already include the varied action outputs that contribute to exploration and the backpropagation is done with respect to this loss and parameters. I would understand if the noise layer has a learned noise parameters in some shape or form, but they do not. We have a simple normal distribution and a decaying/defined standard deviation. This ends up being exactly the same as adding the noise after the output. The noise is not accounted for in the training nor explicitly influences the training pipeline in any way. Moreover, the way it is implemented does not define the turning off of the noise for the actor policy update. If the noise is there during the actor loss calculation (inverse of maximal Q value), you are introducing a stochasticity in a deterministic model. What happens is that you are not evaluating a policy, but rather a noisy approximation of it. This would be analogous to adding random noise to a loss function, and I believe would actually be to a detriment to learning the policy itself (since the loss here is faulty). It most likely works here, because in early steps the policy is very much uncertain, but as you train the introduced noise becomes low enough compared to the total loss here. This method clearly needs more investigation and theoretical as well as practical backing. Moreover, they would need to be compared to standard TD3 baseline without it with classical noise infusion. It would most likely perform the same. Moreover, the response claims that "where both the mean and variance are optimized via backpropagation alongside the policy parameters". But this is not happening the way that it is described in the paper. Backpropagation does not contribute here in any way nor is there any signal for that (See entropy in loss in SAC).
Second. The TD3 purpose description is faulty. The authors claim that the TD3 input space is a 2D lidar with some additional information. Unfortunately, the information in the paper what this additional sensor input is does not agree with the authors response. In any case, what is gathered in every single step is an obstructed 2d map view of the environment for each timestep. Authors claim that the TD3 output is a global reference trajectory, but also claim that the input space is a local (very local if 2d lidar range is 3.5 m) observation. That means that no global plan can be created for the observation, nor global trajectory provided. TD3 can only produce some local trajectory for such limited information. However, here the other issue is that was not addressed by the authors, that what the TD3 outputs is a reactive local navigation signal in a form of linear and angular velocities. If you could roll out the model for multiple future steps, then you would obtain a trajectory, but that does not happen neither in training or deployment. For this you would have to have some sort of a simulator that could forward propagate the states to provide a global trajectory that then could be updated by MPC. It is really unclear what is done here with the TD3 and more specifically how could you get a reference trajectory in deployment. There is a reason why in fig1 there is an image of a global observer, specifically to provide this initial global trajectory, but from local observations and sequential outputs in real-time with TD3 this is very doubtful. Btw, the Fig.1 still mentions DQN even though author response says this was changed.
I find it doubtful that figure 5 has the results over multiple runs with multiple seeds. This looks more like a single run with smoothing enabled. If it would be multiple runs, the first steps of all of these methods would have a standard deviation. But if that would be the case, this would still be not an appropriate design since your training episodes would differ and comparing scores of different episodes would not be appropriate. Additionally, the authors have not responded regarding even comparing to a noise layer in a critic network. It is the same issue as aforementioned noisy loss.
Author Response
We thank the reviewers for their valuable comments.
The manuscript has been revised accordingly and all changes are highlighted.
A detailed point-by-point response is provided in the attached file.

Reviewer 2 Report
Comments and Suggestions for Authors
Thanks to the authors for considering my comments 2-7 and making appropriate changes in the revision. Regarding comment 1, it will be up to the editor to decide if the topic falls within the scope of this Special Issue on `Advances in Bionic Tactile Systems for Sensor Application and Flexible Operation`.
Author Response
We sincerely thank Reviewer 2 for the positive evaluation.
The manuscript has been revised accordingly, and the relevance to the Special Issue has been emphasized.
Detailed responses are provided in the attached file.
